# Endoscopic Hyperspectral Imaging System to Discriminate Tissue Characteristics in Tissue Phantom and Orthotopic Mouse Pancreatic Tumor Model

**DOI:** 10.3390/bioengineering11030208

**Published:** 2024-02-23

**Authors:** Na Eun Mun, Thi Kim Chi Tran, Dong Hui Park, Jin Hee Im, Jae Il Park, Thanh Dat Le, Young Jin Moon, Seong-Young Kwon, Su Woong Yoo

**Affiliations:** 1Department of Nuclear Medicine, Chonnam National University Medical School and Hwasun Hospital, Hwasun-gun 58128, Republic of Korea; 2Institute for Molecular Imaging and Theranostics, Chonnam National University Medical School, Hwasun-gun 58128, Republic of Korea; chichan.1308.edu@gmail.com (T.K.C.T.);; 3Biomedical Science Graduate Program, Chonnam National University, Hwasun-gun 58128, Republic of Korea; 4Department of Artificial Intelligence Convergence, Chonnam National University, Gwangju 61186, Republic of Korea

**Keywords:** hyperspectral imaging, endoscopic imaging, orthotopic mouse model, pancreatic cancer, machine learning

## Abstract

In this study, we developed an endoscopic hyperspectral imaging (eHSI) system and evaluated its performance in analyzing tissues within tissue phantoms and orthotopic mouse pancreatic tumor models. Our custom-built eHSI system incorporated a liquid crystal tunable filter. To assess its tissue discrimination capabilities, we acquired images of tissue phantoms, distinguishing between fat and muscle regions. The system underwent supervised training using labeled samples, and this classification model was then applied to other tissue phantom images for evaluation. In the tissue phantom experiment, the eHSI effectively differentiated muscle from fat and background tissues. The precision scores regarding fat tissue classification were 98.3% for the support vector machine, 97.7% for the neural network, and 96.0% with a light gradient-boosting machine algorithm, respectively. Furthermore, we applied the eHSI system to identify tumors within an orthotopic mouse pancreatic tumor model. The F-score of each pancreatic tumor-bearing model reached 73.1% for the KPC tumor model and 63.1% for the Pan02 tumor models. The refined imaging conditions and optimization of the fine-tuning of classification algorithms enhance the versatility and diagnostic efficacy of eHSI in biomedical applications.

## 1. Introduction

Pancreatic cancer remains the fourth most common cause of cancer-related death, with incidence rates increasing by 1% per year and death rates increasing by 0.2% per year in both sexes [1]. Only 13% of pancreatic cancers are detected at the local stage upon diagnosis when other major regional (29%) and distant metastases (51%) are detected [2]. Due to the advanced stage of the cancer upon diagnosis, only 15% to 20% of patients are candidates for pancreatectomy, which is the only curative method. In addition, despite surgical resection, prognosis is poor. Even in patients with negative margin resections with presumed curative intent, up to 80% can experience disease recurrence [3].

Despite surgical interventions, the high recurrence rates post-surgery highlight the need for more refined imaging modalities to enhance tumor detection and guide therapeutic strategies. Fluorescence-guided surgery (FGS) exhibits promise in tumor detection in the intraoperative period [4,5], and its their reliance on exogenous targeting materials presents limitations in terms of clinical applicability. Thus, the quest for imaging methods devoid of exogenous dyes is paramount for translational research in pancreatic cancer diagnostics.

Hyperspectral imaging (HSI) emerges as a promising candidate in this pursuit, offering the advantages of label-free imaging and ease of assembly in imaging systems [6,7] This modality operates on the principle of capturing a spectral wavelength of light in each pixel, with the help of an element like diffraction grating or a prism [8]. Currently, HSI is applied to many kinds of disease, but it is majorly confined to superficial organs due to the light’s limited ability to penetrate the tissue [9,10].

Recent endeavors have explored the application of hyperspectral imaging in endoscopic settings [11,12]. The endoscopic HSI (eHSI) technique allows us to discriminate neoplastic tissues from normal tissues in the esophagus and colon. However, these applications have predominantly focused on ex vivo tissues, limiting their potential for in vivo evaluations of deep-seated organs. More recently, the first-in-human pilot study of eHSI with a fiber-bundle-based endoscope was applied to the human colon [13]. However, the image quality was limited due to the limitation of the numbers of fiber bundles.

In response to this gap, our study introduces the development of an in vivo eHSI system designed for label-free tissue discrimination. Through the analysis of our lab-built eHSI system, we aimed to assess tissue identification performance within tissue phantoms. The evaluation involved three distinct machine learning-based tissue classification algorithms to identify an optimal classification model. Furthermore, we sought to extend the applicability of eHSI to an in vivo setting by evaluating its performance in an orthotopic mouse pancreatic tumor model. The outcomes of this study not only contribute insights into tissue discrimination capabilities but also provide a foundation for considering translational research opportunities in the context of pancreatic cancer diagnostics.

## 2. Materials and Methods

### 2.1. Development of the eHSI System

The eHSI system developed for this study incorporated a liquid crystal tunable filter (LCTF) to enable spectral scanning (Figure 1). Our eHSI system featured a dual-camera setup to enhance imaging capabilities. The rigid endoscope (HOPKINS II Telescope 27301AA, Karl Storz, Tuttlingen, Germany), strategically positioned at the front of the system, facilitated efficient data collection. A high-power light-emitting diode (LED) light source (TouchBright X6, Live Cell Instrument, Seoul, Republic of Korea) was applied as an illumination light source. This configuration allowed for the simultaneous acquisition of white light endoscopic color images and hyperspectral data. An achromatic lens (focal length = 75 mm, LA1608-A-ML, Thorlabs, NJ, USA) and a beam splitter (30:70 (reflection– transmission)), BS049, Thorlabs, NJ, USA) were assembled between the endoscope and detection cameras to obtain images from the white-light and HSI cameras simultaneously. A high-quality color CMOS camera achieved white light endoscopic color imaging (acA1920-40uc, BASLER AG, Ahrensburg, Germany). The white light images served as a reference for comparison with the hyperspectral data, aiding in the interpretation and analysis of tissue characteristics.

Hyperspectral imaging using the eHSI system was carried out with a scientific CCD camera (Retiga R1, Qimaging, Surrey, BC, Canada) connected to the LCTF. The LCTF utilized in our custom-built system was the Kurios-WB1/M model from THORLABS (NJ, USA). This component was crucial in capturing hyperspectral information from the target tissues. In addition, all components were attached on the aluminum breadboard (MB3030/M, Thorlabs, NJ, USA) and articulated on the ball stage (SL20/M, Thorlabs, NJ, USA) to make a better position. The field of view (FOV) and spatial resolution were assessed at a working distance of 10 mm using the USAF 1951 resolution target (R1L1S1P, Thorlabs, NJ, USA). The FOV measured 11.1 mm, while the spatial resolution was determined to be 18.0 line pairs per millimeter (lp/mm).

The imaging workflow involved the coordinated functioning of the dual-camera system. While the white light endoscopic color images provided a baseline for anatomical reference, the eHSI data acquisition process involved scanning the target tissues using the LCTF and attaching the CCD camera behind the LCTF. The hyperspectral images captured by the scientific CCD camera facilitated the extraction of spectral signatures for subsequent analysis. 

### 2.2. Tissue Phantom Imaging with the eHSI System

Fresh pork tissue phantoms were employed to assess the performance of the developed eHSI system. The exposure time of CCD camera for the eHSI system imaging was set at 500 ms for each wavelength, and the spectral range spanned from 420 nm to 730 nm with an imaging wavelength interval of 10 nm. The light source intensity was maintained at 100%, ensuring consistent illumination during the imaging process.

Simultaneously, white light endoscopic color images were acquired using a color CMOS camera with an exposure time of 30 ms, corresponding to video rate conditions. This parallel acquisition of white light images provided a real-time reference for anatomical features, complementing the hyperspectral data obtained by the eHSI system.

Following image acquisition, each wavelength-specific image generated by the eHSI system was saved in *.tif file format.

### 2.3. Image Analysis

The analysis of hyperspectral data obtained from the tissue phantom imaging experiments was conducted through a systematic and comprehensive approach. The raw *.tif images, representing different wavelengths, were merged into a single ENVI file format using MATLAB ver. R2022b (Mathworks, Inc., Natick, MA, USA) This consolidation facilitated efficient handling and processing of the hyperspectral dataset.

To conduct the analysis, we utilized the commercial software Breeze (Prediktera AB, Umea, Sweden). Backgrounds were eliminated from individual images using pseudo RGB representations derived from the hypercube within the software. Spectral preprocessing procedures included principal component analysis (PCA) alongside standard normal variate (SNV) transformation and mean centering to mitigate the effects of light scattering. Test samples underwent manual annotation after delineating three regions of interest (ROIs) corresponding to known tissue characteristics, such as fat or muscle (Figure 2). Within each ROI, 500 representative pixels were evenly selected to construct representative spectra showcasing the tissue characteristics.

For the training phase of tissue identification, we selected the light gradient boosting machine (LGBM) algorithm following an automated evaluation of macro accuracy across various available algorithms within the software, including decision trees, random forests, support vector machines, maximum entropy, and logistic regression. The training of these algorithms utilized ten pre-labeled eHSI datasets.

Upon completion of the training phase, the discrimination of tissue areas was executed on the single test sample. In this stage, three distinct algorithms were deployed: support vector machine (SVM), neural network (NN), and light gradient boosting machine (LGBM). Subsequently, the pre-trained classification model was applied to additional single hyperspectral images obtained from different tissue phantom sites to assess the system’s ability to discern between fat and muscle tissues. The utilization of three different algorithms facilitated the examination of performance disparities and identification of the most effective approach for tissue classification.

The evaluation of classification performance employed precision, recall, and F-score metrics, which were specifically selected to gauge the accuracy of the classification model in distinguishing fat tissue from muscle. The adoption of multiple algorithms and rigorous evaluation metrics contributes to a comprehensive understanding of the capabilities and constraints of the developed eHSI system in discerning tissue characteristics within the tissue phantom model.

### 2.4. Generation of Orthotopic Mouse Pancreatic Tumor Model

This animal study was conducted in compliance with ethical standards and received approval from our Institutional Animal Care and Use Committee (IACUC). Six-week-old female C57BL/6 mice were purchased for the study (Gbio, Gwangju, Republic of Korea). The mice underwent a preparatory phase that involved the removal of hair from the abdominal area using a combination of a hair remover machine and hair removing cream. Subsequently, a carefully executed incision was made in the lateral ventral region of the mouse, exposing the abdominal cavity, and we proceeded to make an orthotopic pancreatic tumor model (Figure 3).

The pancreas was identified using forceps, and mouse pancreatic cancer cells (KPC, 1 × 10^5^/20 µL; Pan02, 3 × 10^5^/20 µL) were injected directly into the pancreatic tissue via an insulin syringe (Ultra-fineTM II, BD, Biosciences, San Jose, CA, USA). To prevent the reflux of cancer cells into the abdominal cavity, the injection site was ligated using vicryl sutures. Following the injection, the abdominal wall and skin were sutured to complete the surgical procedure. After a two-week period, tumor formation was assessed by making a small incision in the abdominal wall. The presence and characteristics of the tumors were examined, ensuring that the model accurately represented pancreatic tumor development.

### 2.5. eHSI in Orthotopic Pancreatic Tumors

The developed eHSI system was strategically employed in the assessment of the orthotopic mouse pancreatic tumor model (Figure 4). The primary objective was to leverage the advanced imaging capabilities of the eHSI system to discriminate between pancreatic tumors and adjacent normal tissues. The imaging conditions for the eHSI system during this application were configured. The exposure time for each wavelength was set at 500 ms, spanning a wavelength range of 420–730 nm with a 10 nm imaging wavelength interval. The light source intensity was adjusted to 60% to optimize imaging conditions. In addition, the color camera image served as a guide to align the imaging field of the eHSI system, ensuring precise targeting of the pancreatic tumor and surrounding tissues.

Similar to the image analysis process applied in the previous tissue phantom experiments, the obtained hyperspectral images were subjected to analysis using Breeze software ver. 2023.2.0 (Prediktera AB, Umea, Sweden). The algorithm training was carried out with single pre-labeled eHSI obtained from orthotopic pancreatic tumor model. The pre-trained classification model was then applied to other single eHSI obtained from a different orthotopic pancreatic tumor model, allowing for the evaluation of the system’s ability to discriminate tumors from other normal tissues. The tumor classification performance of each algorithm was expressed as precision, recall, and F-score.

## 3. Results

### 3.1. Tissue Classification Performance of eHSI Images in Tissue Phantoms

The tissue classification analysis with eHSI showed notable ability to discriminate between fat and muscle tissue (Figure 5). The classification software marked the fat tissue of the tissue phantom with a clear margin.

All three machine learning algorithms showed strong tissue classification performance in discriminating fat from muscle in a tissue phantom (Table 1). Regarding precision, SVM, NN, and LGBM achieved 98.3%, 97.7%, and 96.0%, respectively. Additionally, all three classification models demonstrated commendable recall values, with SVM achieving 93.4%, NN 93.8%, and LGBM 97.2%. The overall classification performance measured by the F-score, with SVM leading at 95.8%, followed closely by NN at 95.7% and LGBM at 96.6%. Notably, SVM exhibited the highest precision among the algorithms, showcasing its efficacy in discriminating fat tissue from muscle within the tissue phantom model.

### 3.2. Tissue Classification Performance of eHSI in Orthotopic Pancreatic Tumors

Hematoxylin and eosin staining was applied to the extracted pancreas from the control (without cancer cell implantation), KPC, and Pan02 tumor-bearing mouse models (Figure 6). Compared with the control group, bulky tumor formation was observed within the pancreas in the KPC and Pan02 tumor-cell-implanted groups. Magnified tissue images showed tumor cells located within the normal pancreatic cells.

The outcomes of the classification analysis with eHSI data, elucidated in the presented tables, underscore the effectiveness of three distinct machine learning algorithms—SVM, NN, and LGBM—in distinguishing tumor tissues within the examined models (Table 2). 

In the KPC tumor-bearing model, SVM exhibited a substantial precision of 91.5%, closely followed by NN at 91.7%, and LGBM at 89.6%. Similarly, within the Pan02 tumor-bearing model, SVM demonstrated a precision of 83.0%, surpassing NN at 82.5% and LGBM at 73.3%. A representative image with NN is presented in Figure 7.

Regarding recall values, SVM displayed a recall of 58.5%, while NN and LGBM achieved 57.9% and 61.7%, respectively, in the KPC tumor-bearing model. In the Pan02 model, SVM exhibited a recall of 50.9%, exceeding NN at 50.8% and LGBM at 48.8%.

Furthermore, the F-score, serving as the harmonic mean of precision and recall, represents a balanced metric for evaluating the efficacy of tumor classification. In the KPC model, SVM yielded an F-score of 71.4%, closely followed by NN at 71.0% and LGBM at 73.1%. Similarly, within the Pan02 model, SVM attained an F-score of 63.1%, outperforming NN at 62.9% and LGBM at 58.6%. These metrics collectively underscore the nuanced performance of the algorithms in achieving a harmonious balance between precision and recall in the classification of tumor tissues.

## 4. Discussions and Conclusions

Our results present the development and evaluation of an eHSI system for discriminating tissue characteristics. The eHSI system exhibited robust performance in tissue phantom imaging, successfully distinguishing between fat and muscle tissues. Subsequently, the system was applied to an in vivo orthotopic pancreatic tumor model, employing three machine learning-based classification algorithms. While the results demonstrated commendable precision, the recall performance was comparatively limited, highlighting the nuanced challenges in classifying tumor tissues within a biological context.

The fundamental basis for disease diagnosis employing his lies in the alterations of tissue optical properties induced by morphological and biochemical changes associated with disease progression [14]. These modifications empowhisHSI to discern lesions and abnormal tissue, obviating the need for histological examination and thereby enhancing efficiency and treatment outcomes. Reflectance, absorption, and scattering phenomena spanning the visible to near-infrared light spectrum (400–1000 nm) furnish diagnostic insights into tissue physiology, morphology, and composition [15]. Each wavelength of light interacts uniquely with the material, influenced by its chemical composition, including water content, hemoglobin levels, lipid concentrations, and other molecular constituents. For instance, in pancreatic cancer, enlarged nuclear size in adenocarcinoma cells alters the optical signature of reflectance spectra compared to normal pancreatic tissue [16]. HSI can detect these changes and aid in discriminating tumors from normal tissues.

Morehisr, HSI finds application in disease monitoring during anticancer therapy by reflecting tissue perfusion and oxygenation dynamics [17,18]. When anti-angiogenic drugs are administered in breasthisncer, HSI effectively captures changes in oxygenation, reflecting tumor responses [19]. This underscores the phisntial of HSI as a label-free monitoring modality, garnering attention for its non-invasive and real-time assessment of disease dynamics.

The pursuit of employing eHSI for cancer detection dates back to 2004 [20]. Initial attempts involved acquiring eHSI data on lung epithelial tissue, marking the inception of broader research endeavors into lung cancer detection. Subsequently, the application of eHSI expanded to encompass diverse biomedical approaches. However, the inherent challenge lies in the time-intensive nature of multiple imaging procedures, limiting real-time applications. In response, some researchers have adopted a pragmatic approach, obtaining hyperspectral images with a restricted number of spectral bands (less than six) [21,22]. Alternatively, others have sought to identify singular wavelengths that exhibit the most substantial contrast between normal and neoplastic lesions [23]. Despite promising outcomes reported in the literature, the efficacy of hyperspectral imaging (HSI) was constrained by the limited number of spectral bands.

To address this limitation, recent efforts have applied a more extensive spectral band approach in eHSI applications for laryngeal [24] and upper gastroesophageal malignancies [11,25]. The discrimination between tumors and adjacent normal tissues has been accomplished through the deployment of classification methods. Presently, a transition towards the clinical translation of eHSI is occurring; this is being facilitated by the commercialization of eHSI systems, with the aim of assessing tissue perfusion [26] and identifying key anatomical structures, such as blood vessels and nerves, during surgical procedures [27]. In line with these investigations, our study amalgamated this approach with machine learning algorithms for in vivo pancreatic tumor analysis.

To attain superior performance in tissue classification, we employed three machine learning-based classification algorithms: SVM, NN, and LGBM. In the tissue phantom, all three algorithms showed a higher F-score, with the highest value in LGBM (96.6%). Meanwhile, in the orthotopic pancreatic tumor model, the overall F-score was highest in LGBM (73.1%) in the KPC tumor-bearing mouse model and in SVM (63.1%) in the Pan02 tumor-bearing mouse model. 

SVM, renowned for its robustness and efficacy, stands as one of the most widely utilized predictors in classification problem domains. It is designed to ascertain the optimal hyperplane for the separation of data, a process integral to the classification of diverse classes [28,29]. An illustrative application of SVM in the realm of medical imaging is evident in the work of Urbanos G. et al., who utilized SVM to discriminate brain tumors from adjacent healthy tissues, achieving an impressive overall accuracy of 76% [30].

The artificial NN model exhibits favorable attributes such as simplicity and robustness [31]. The architecture of the network involves source nodes representing the input layer, a single hidden layer of computation nodes, and an output layer comprising two nodes. This configuration of the ANN is well suited to addressing multi-dimensional mapping challenges, particularly when provided with consistently structured data and a sufficient number of neurons within its hidden layer. Demonstrating the efficacy of this approach, Halicek M. et al. applied a convolutional NN for the classification of tumor margins in head and neck cancer, achieving a notable accuracy of 90% in the detection of thyroid carcinoma margins [32].

The LGBM represents an enhanced iteration of a gradient-boosting machine (GBM), leveraging tree-based learning techniques. Its notable capabilities include the adept handling of extensive datasets and the achievement of high-accuracy outcomes while operating within constraints of computing resources, such as memory space and computing speed. This proficiency positions LGBM as a favorable model when compared to alternative approaches [33]. Kim H. et al. used LGBM to classify early-stage laryngeal cancer by classification of voice change. The sensitivity, specificity, and accuracy rates were 70.0%, 73.3%, and 71.5%, respectively [34].

Several limitations warrant consideration in our study. Firstly, a relatively small number of samples were utilized, and we did not conduct statistical significance evaluations among the machine learning algorithms. However, each region of interest (ROI) was subdivided into 500 pixels, enabling a detailed assessment of 1500 spectral features for each tissue. Further improvements in tissue classification performance may be attainable through the utilization of a more extensive dataset. Secondly, significant variations were observed in the classification performance between the tissue phantom and mouse tumor model. The limited sample size makes it challenging to interpret these variations, which may stem from differences in tissue structures, tumor morphology, and physiological factors. The in vivo conditions also introduced challenges, such as light reflections from moist organs, necessitating additional image processing to mitigate these confounding factors. Employing more sophisticated approaches with tissue condition compensation methods will be necessary. Thirdly, we employed a limited number of classification algorithms that are commonly used in tissue classification. Future investigations should involve larger-scale image datasets and comprehensive comparisons of algorithms to identify the optimal approach for tissue classification.

In conclusion, our developed eHSI system exhibits promising applications for discriminating tissue characteristics in both tissue phantoms and in vivo mouse tumor models. While the study has provided valuable insights into the system’s capabilities, optimization of imaging conditions and algorithm selection are pivotal for advancing the practicality and diagnostic potential of eHSI in biomedical applications. Future endeavors focusing on these aspects will undoubtedly contribute to the refinement and broader applicability of this innovative imaging modality.

## Figures and Tables

**Figure 1 bioengineering-11-00208-f001:**
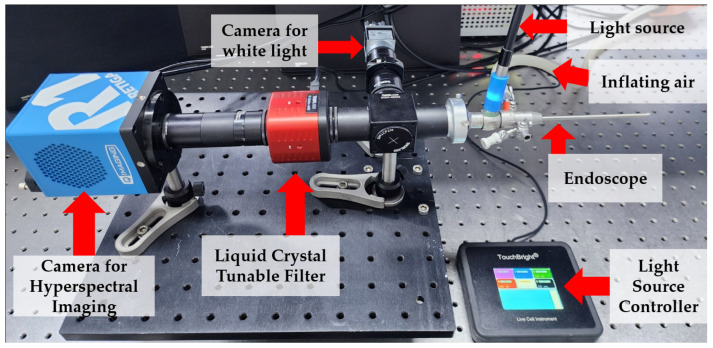
Endoscopic hyperspectral imaging system. A liquid crystal tunable filter was applied between the endoscope and camera to obtain hyperspectral imaging.

**Figure 2 bioengineering-11-00208-f002:**
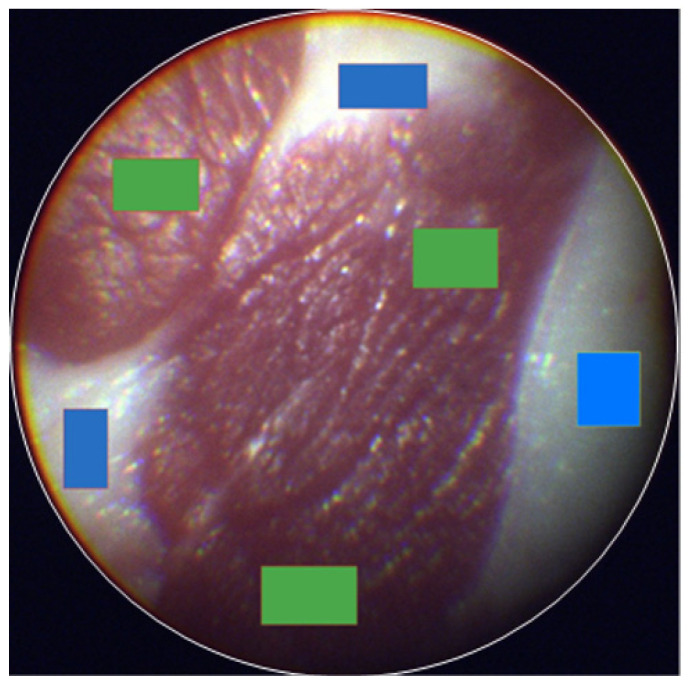
Image classification of eHSI in a tissue phantom. The muscle (green rectangle) and fat (blue rectangle) of tissue phantoms were labeled with multiple rectangular regions of interest (ROIs).

**Figure 3 bioengineering-11-00208-f003:**
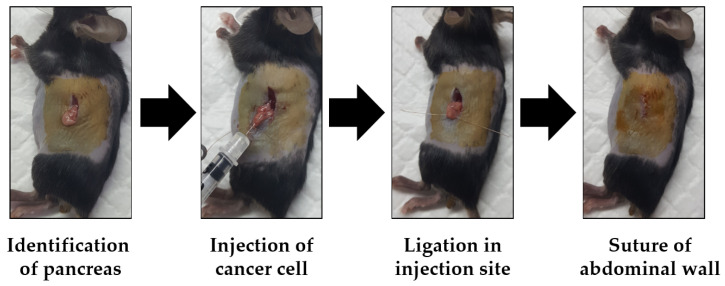
The surgical process of generating the mouse pancreatic tumor model. After identifying the mouse pancreas, pancreatic cancer cells were injected into the pancreatic tissue. A suture was applied to the injection site to prevent accidental spillage of cancer cells. After the suture was finished, the abdominal wall and skin were sutured with surgical suture material.

**Figure 4 bioengineering-11-00208-f004:**
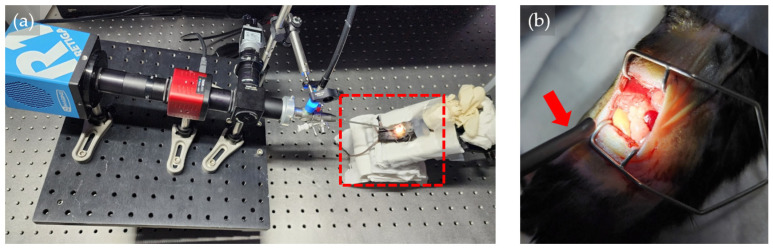
Endoscopic hyperspectral imaging (HSI) system in an orthotopic pancreatic mouse tumor model. (**a**) Tilted endoscopic HSI system focused on the mouse abdomen. (**b**) Magnified images from the red-dotted area in Figure (**a**). Tip of endoscope (red arrow) focused on the open belly of the mouse abdomen.

**Figure 5 bioengineering-11-00208-f005:**
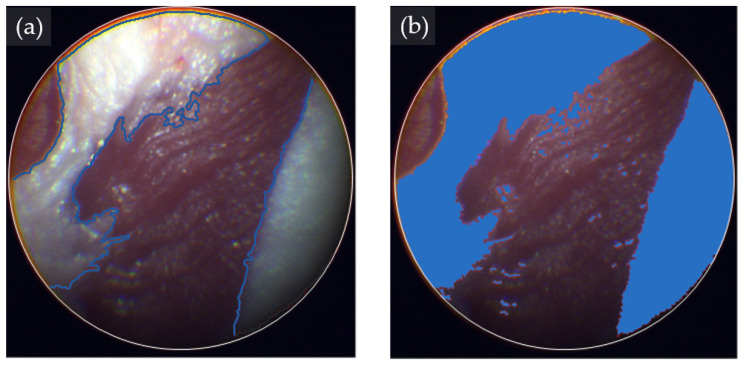
Hyperspectral image (HSI)-based tissue classification in a tissue phantom. (**a**) Pseudo RGB image showing muscle and fat tissues of a tissue phantom. The margins are highlighted with blue lines. (**b**) HSI-based classification of fat tissues (blue-colored area) and muscle tissues using the support vector machine algorithm.

**Figure 6 bioengineering-11-00208-f006:**
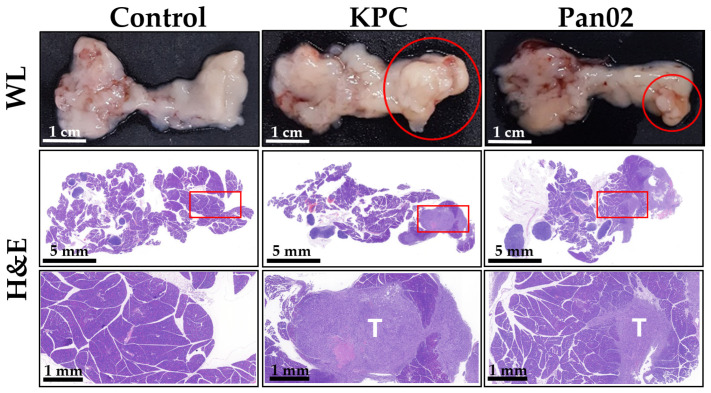
Histological examination of orthotopic pancreatic tumors. The white light (WL) image depicts tumor formation (red circle) within the pancreas of mice implanted with KPC (**middle column**) and Pan02 (**right column**) tumor cells, contrasted with the control group (**left column**). Hematoxylin and eosin (H&E)-stained images, with a magnified view of the delineated red rectangular area, illustrate the presence of tumor cells (T) amidst the glandular architecture of the normal pancreatic tissue.

**Figure 7 bioengineering-11-00208-f007:**
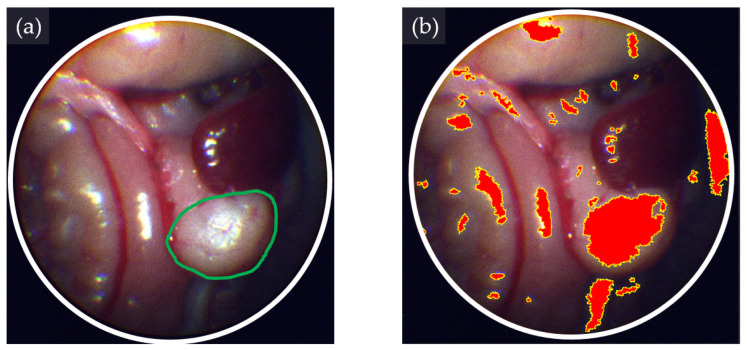
Hyperspectral image (HSI)-based tissue classification in an orthotopic mouse pancreatic tumor model. (**a**) Pseudo RGB image showing a KPC pancreatic tumor-bearing mouse. Tumor margins are highlighted by green lines. (**b**) HSI-based tissue classification of a pancreatic tumor (red-colored area) using the neural network classification algorithm.

**Table 1 bioengineering-11-00208-t001:** Classification performance in discriminating fat from muscle in a tissue phantom according to different artificial intelligence-based training algorithms.

	SVM ^1^	NN ^2^	LGBM ^3^
Precision	98.3%	97.7%	96.0%
Recall	93.4%	93.8%	97.2%
F-score	95.8%	95.7%	96.6%

^1^ SVM, support vector machine; ^2^ NN, neural network; ^3^ LGBM, light gradient boosting machine.

**Table 2 bioengineering-11-00208-t002:** Classification performance in discriminating tumors from adjacent normal tissue in two orthotopic pancreatic tumor models with three different machine learning algorithms.

	KPC Tumor Bearing Model			Pan02 Tumor Bearing Model		
	SVM ^1^	NN ^2^	LGBM ^3^	SVM ^1^	NN ^2^	LGBM ^3^
Precision	91.5%	91.7%	89.6%	83.0%	82.5%	73.3%
Recall	58.5%	57.9%	61.7%	50.9%	50.8%	48.8%
F-score	71.4%	71.0%	73.1%	63.1%	62.9%	58.6%

^1^ SVM, support vector machine; ^2^ NN, neural network; ^3^ LGBM, light gradient boosting machine.

## Data Availability

The datasets used and/or analyzed during the current study are available from the corresponding author on reasonable request.

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
