# Peer review of "Endoscopic Hyperspectral Imaging System to Discriminate Tissue Characteristics in Tissue Phantom and Orthotopic Mouse Pancreatic Tumor Model"

_bioengineering, 2024, doi:10.3390/bioengineering11030208_

Round 1

Reviewer 1 Report

Comments and Suggestions for Authors

Although the paper offers insightful information about endoscopic hyperspectral imaging system to discriminate tissue characteristics in tissue phantom and orthotopic mouse pancreatic tumor model, some important issues are left out. Please include the answers to these questions in the text if applicable.

1.      The eHSI system utilizes an achromatic lens in conjunction with a liquid crystal tunable filter (LCTF) to reduce chromatic aberration. Achromatic lenses are designed to bring two wavelengths (typically red and blue) into focus in the same plane. The LCTF selectively transmits wavelengths, ensuring consistent spectral resolution. The system likely employs advanced optical calibration to ensure uniform spectral response and minimize aberrations across the field of view. How does the endoscope's optical design in the eHSI system mitigate chromatic aberration and ensure uniform spectral resolution across the entire field of view?

2.      The eHSI system likely reveals unique spectral signatures associated with different tissue types based on their biochemical composition and vascularization patterns. Malignant tissues often show altered metabolism and increased blood supply, leading to distinct spectral profiles. Specific characteristics of pancreatic tissue were not detailed in the article. What specific pancreatic tissue characteristics are most distinctly revealed in the eHSI system, and how do these assist in differentiating between benign and malignant lesions?

3.      The article does not specify methods for compensating for light scattering and absorption in deeper tissues. Typically, techniques such as mathematical modeling of light-tissue interaction or applying scattering correction algorithms are used. These corrections are vital for accurate tissue characterization, especially in layers not immediately adjacent to the tissue surface. Was there any compensation or correction method used in the eHSI system to account for light scattering and absorption differences in deeper tissue layers?

4.      The article doesn't mention specific feature extraction techniques used before applying machine learning algorithms. Generally, techniques such as Principal Component Analysis (PCA), Independent Component Analysis (ICA), or spectral angle mapping could be employed to reduce dimensionality and highlight critical features in hyperspectral data for efficient classification. What feature extraction techniques were used to preprocess the hyperspectral data before feeding it into the machine learning algorithms for tissue classification?

5.      The article does not provide specific details about the resolution of the eHSI system compared to traditional histopathology. While eHSI can provide detailed information about tissue composition and structure, it typically does not reach the cellular resolution offered by microscopic histopathological examination. Early neoplastic changes at the cellular level might not be discernible using eHSI alone. How does the eHSI system's resolution compare to traditional histopathological examination, and can it identify cellular-level changes, such as those seen in early neoplastic transformations?

6.      The article does not address the issue of tissue autofluorescence. Autofluorescence, inherent in biological tissues, can be a significant source of noise in hyperspectral imaging. Techniques such as fluorescence lifetime imaging or the use of specific spectral bands that minimize the impact of autofluorescence might be employed, but the article lacks details on this aspect. How does the eHSI system handle the issue of autofluorescence from tissues, which can often interfere with the accurate spectral analysis?

7.      The article doesn't delve into how the eHSI system might be applied to analyze interactions between drugs and tissues, particularly in the context of chemotherapy treatments for pancreatic tumors. This type of analysis would necessitate further experimental work, concentrating on how tissues' spectral characteristics change post-drug administration. Has there been any investigation into the effects of chemotherapeutic agents on pancreatic tumor tissues using the eHSI system, specifically looking at drug-tissue interactions?

Author Response

Dear Expert reviewers,

Thank you for coordination the review of our manuscript. We also thank the reviewers for their comprehensive and critical evaluations. We have addressed the reviewer’s comments and made substantial and appropriate revisions to the manuscript as requested. There revisions have led to the inclusion of text and references in the original manuscript. The revised parts have been highlighted in the manuscript.

Please see the attached file. Thank you.

Reviewer 2 Report

Comments and Suggestions for Authors

This work reports the study of an endoscopic hyperspectral imaging system for discriminating tissue characteristics in tissue phantom and orthotopic mouse pancreatic tumor model.

Similar in vivo investigations of using an endoscopic hyperspectral imaging system for analysis have been reported in the literature. For example, Surg. Endosc. 37, 3691–3700, 2023; J. Biophotonics, 15(3), e202100167, 2022; Diagnostics 11(8):1508, 2021; Cancers 11(6):756, 2019 (I am not in any way related to those authors, or journals). Please note that these are just few examples and more could be found. Hence, please highlight more about the significant novelty/advantage, not just minor dissimilarity, in your work when compared with other similar works reported in the literature. This emphasis could help general journal readers to understand the novelty in your work.

Please justify the large variations in values for recall and f-score between phantom and mouse model. Both values are in the 0.9 range for tissue phantom, while values for recall and f-score are about 0.5 and 0.7, respectively for KPC/Pan02 tumor model (regardless of the machine language used). Moreover, please comment on high recall values in the tissue phantom, since these values are related to false negative, which is of great importance and grave consequences in medical diagnosis.  Additionally, please also justify the discrepancies in trends for various machine languages. For example, in table 2, neural network has the highest value in precision for KPC model, but not in Pan02 model; LGBM has the highest value in recall/f-score for KPC model, but lowest in Pan02 model.  

Please provide the number of samples measured/analyzed, and standard deviations for values. Please justify the reason for not performing repeatability tests, or analysis for other region of interest within the same sample (if these were not performed).

For H&E imagines in Figure 6, the scale bar seems to be incorrect. For example, 5x magnification has an enlargement more than 5x when comparing features in the middle column. Please explain.

More quantitative comparisons and discussions could have been done in terms of precision, recall and f-score with others techniques in the literature. Understand that dissimilar conditions and parameters in different situations can be difficult to compare, yet, this comparison gives a rough idea to general readers where your strategy stands in various aspects (I am not trying to say which strategy is better, each strategy has its own advantages and weaknesses in different areas).

Author Response

Dear Expert reviewers,

We thank the reviewers for their comprehensive and critical evaluations. We have addressed the reviewer’s comments and made substantial and appropriate revisions to the manuscript as requested. There revisions have led to the inclusion of text and references in the original manuscript. The revised parts have been highlighted in the manuscript.

Please see the attached file. Thank you.

Round 2

Reviewer 1 Report

Comments and Suggestions for Authors

I'm pleased to let you know that after a careful review, your manuscript is now in excellent condition for publication. All of the earlier concerns have been successfully addressed by the revisions, greatly improving the paper's overall quality and clarity. Your research's breadth and depth are noteworthy for their value and depth, contributing notably to the field.

I appreciate your commitment to polishing the manuscript. I do not doubt that the academic community will value your work and that it will significantly advance current discussions and research in your area.

Reviewer 2 Report

Comments and Suggestions for Authors

Thank authors for answering my comments and revising accordingly in the manuscript.